# Gene silencing for invasive paper wasp management: Synthesized dsRNA can modify gene expression but did not affect mortality

Mariana Bulgarella ⬡ �} *, James W. Baty �} , Rose McGruddy, Philip J. Lester

School of Biological Sciences, Victoria University of Wellington, Wellington, New Zealand

} These authors contributed equally to this work.
* Mariana.Bulgarella@vuw.ac.nz

**Data Availability Statement:** All relevant data are within the paper and its Supporting Information files.

## Abstract

Invasive paper wasps such as *Polistes dominula* are a major pest and problem for biodiversity around the globe. Safe and highly targeted methods for the control of these and other social wasp populations are needed. We attempted to identify potentially-lethal gene targets that could be used on adult paper wasps in a gene silencing or RNA interference (RNAi) approach. Double-stranded RNA (dsRNA) was designed to target genes for which silencing has proven lethal in other insects. dsRNA was provided either orally to foragers or directly injected into the wasps. We also provided the dsRNA unprotected or protected from degradation by gut nucleases in two different forms (lipofectamine and carbon quantum dots). The effects of oral delivery of 22 different gene targets to forager wasps was evaluated. The expression of five different genes was successfully reduced following dsRNA ingestion or injection. These gene targets included the FACT complex subunit spt16 (DRE4) and RNA-binding protein fusilli (FUSILLI), both of which have been previously shown to have potential as lethal targets for pest control in other insects. However, we found no evidence of significant increases in adult wasp mortality following ingestion or injection of dsRNA for these genes when compared with control treatments in our experiments. The methods we used to protect the dsRNA from digestive degradation altered gene expression but similarly did not influence wasp mortality. Our results indicate that while many of the same gene targets can be silenced and induce mortality in other insects, dsRNA and RNAi approaches may not be useful for paper wasp control.

## Introduction

RNA interference (RNAi) technology is a promising, environmentally-friendly method to control insect populations that uses double stranded RNA (dsRNA) or small interfering RNA (siRNA) to trigger post-transcriptional gene silencing [1]. The experimental approach of RNAi-mediated gene targeting involves the capacity of cells to degrade target messenger RNA (mRNA) with sequence homology to the administered dsRNA [2]. The intracellular RNAi pathway includes the Dicer enzyme which cleaves dsRNA into siRNAs that are then loaded

**Funding:** This study was funded by a Smart Ideas grant number PROP-61270-ENDSI-RSCHTRUSTVIC from the New Zealand Ministry of Business, Innovation and Employment, awarded to PJL. https://www.mbie.govt.nz/science-and-technology/science-and-innovation/funding-information-and-opportunities/investment-funds/endeavour-fund/application-and-assessment-information/ The funders had no role in study design, data collection and analysis, decision to publish, or preparation of the manuscript.

**Competing interests:** The authors have declared that no competing interests exist.

into RNA-induced silencing complexes (RISC). The siRNA fragments guide the RISC to target complementary mRNA for degradation [3]. RNAi is considered an almost universal method of gene silencing [4].

The main benefit of the RNAi approach is its specificity, where the dsRNA sequence is designed specifically for the pest of interest, capable of suppressing genes critical for growth, development or reproduction in that target species only. This is unique in the sense that RNAi is selectively targeting the species of interest without having non-target effects on other species [5]. The main challenge in widespread use of this technology is the development of inexpensive and reliable dsRNA production and delivery methods [6]. Modes of delivery of the RNAi target to insects include soaking, feeding or microinjection, or even genetically modifying plants or bacteria to produce dsRNA fragments [7]. One of the most convenient ways to deliver RNAi for insect control in the field is orally. However, response to ingested dsRNA varies greatly in insects with beetles being highly susceptible to it whereas dsRNA effects are variable in moths, butterflies, flies, mosquitoes, aphids, hoppers and stinkbugs, for example [reviewed in 8].

The varying efficiency of RNAi technology observed in different insects depends largely on several mechanisms including dsRNA stability during or after entering the insect, insufficient dsRNA internalization, deficient RNA machinery and impaired systemic spreading, among others [3, 8]. Once dsRNA is consumed by the insect, it must avoid degradation by ribonucleases from salivary glands, midgut and the hemolymph [8]. Gut ribonucleases are a major bottleneck decreasing the efficiency of RNAi in insects, being responsible for degradation of orally-ingested dsRNA [9]. There are different methods of protecting the dsRNA against the ribonucleases in the insects' digestive system. Protection methods include silencing ribonuclease genes or conjugating the applied dsRNA with nanoparticles [9]. At least four nanoparticles are known to aid in the delivery of dsRNA; Carbon Quantum Dots (CQDs), chitosan, amine functionalized silica nanoparticle, and lipofectamine reagent. Previous studies compared the efficiency of three of these nanoparticles and found that CQDs were the most efficient carrier for dsRNA retention and delivery therefore increasing gene silencing and mortality in mosquitos [10] and the rice striped stem borer [11].

Invasive insects are a worldwide problem whose consequences affect not only the economy but the ecosystem via competition and/or predation upon native wildlife. Hymenoptera, in particular, have successfully invaded many regions of the globe and have become pests. Several worldwide research groups are focusing their efforts on control methods for social insects. For example, in Florida, USA, wasps aggregated in large numbers on shuttle pads of the National Aeronautics and Space Administration (NASA). *Polistes* males swarm at elevated man-made structures and towers waiting for females to arrive. After mating, the females remain in these structures forming hibernating clusters from which they disperse in the spring [12, 13]. This wasp presence is a nuisance for workers and equipment alike. Consequently, the United States Air Force began funding research on potential control methods for *Polistes* wasps [12].

Currently there are no effective control methods available exclusively for paper wasps, other than the direct application of insecticides to these wasps or their nests [14]. Effective and highly targeted control methods are needed considering these wasps are invasive and a costly pest in many continents [15–19]. Attractants associated with fermented food are used in lures with varying degrees of efficacy [20, 21]. Generic insecticide sprays and wasp baits are the only control methods currently available [14]. No study to date has determined the potential of dsRNA feeding as a lethal control method for *Polistes* paper wasps. We are aware of two studies on dsRNA feeding and efficacy in *Polistes* paper wasps [22, 23]. In both these publications, the objective was to determine the effect of knocking down genes thought to be involved in caste differentiation in field experiments [22, 23], and were not aimed at developing a control method.

In this study our aim was to determine if candidate genes could be identified, which if their expression could be reduced by targeted dsRNA, would be lethal for adult *Polistes dominula*. The gene targets we investigated were based on those found to be lethal in other insect pest species. Wasp foragers were injected dsRNA directly, or workers and even entire nests were fed dsRNA in captivity. The ultimate goal of our study was to contribute towards the development of a species-specific control option for large spatial scale use in the field in areas where this species is a widely distributed pest.

## Materials and methods

### Wasp collection and captive rearing

We collected *Polistes dominula* colonies in the Nelson surrounding areas, South Island, New Zealand (41.2708˚S, 173.1208˚E) during the Austral summers of 2019, 2020 and 2021. Collection or research permits were not needed for this study because *P. dominula* is an exotic species that is not listed under a governmental pest management plan either for the collection site (Nelson) or in Wellington where the experiments took place. Colonies were collected in individual bags and transferred to plastic containers for transport to a quarantine laboratory at Victoria University of Wellington. The nests were placed in wire mesh cages (60 x 60 x 60 cm, BioQuip Products, California, USA) in the laboratory. Wasps were kept in a rearing room at a temperature of 25 ± 1˚C, relative humidity of 40 ± 10%, and Light/Dark photoperiod of 14:10 h. They were provided *ad libitum* with 1.75 mol/L raw sugar water (Chelsea, New Zealand), fondant bee candy (Apifeed, New Zealand), drinking water and wax moth (*Galleria mellonella*) larvae (Biosuppliers Ltd., New Zealand) as a protein source. Initially the wasps did not appear to recognise the larvae as food, so we needed to train the wasps to recognise the wax moth larvae as a prey item. Training was achieved by placing injured larvae in petri dishes directly (<5 cm) underneath the nests, which appeared to initiate an aggressive and defensive response from the adult wasps on the nest. They would attack the larvae and appeared to then discover they were food. Larvae and water were replenished every other day. Each cage was enhanced with artificial aquarium plants to provide shelter from female aggression for males [following 24] and small tree branches for the wasps to perch on. Building cardboard and stripped paper was provided *ad libitum* in each cage.

### Candidate gene ortholog identification

Potentially lethal candidate genes were selected according to previous dsRNA research based on pest insects such as *Tribolium castaneum*, *Diabrotica virgifera virgifera* [25–27], *Blatella germanica* [28], *Anthonomus grandis* [29], *Varroa destructor* and *V. jacobsoni* [30, 31]. Candidate genes were identified with BLASTn searches in *P. dominula*'s genome against the sequences of *T. castaneum*, *D. virgifera*, *B. germanica*, *A. grandis and V. destructor* in NCBI GenBank [32]. dsRNA was manufactured by RNA Greentech LLC (Texas, USA). The dsRNA was synthesized using the *in vitro* transcription method, which requires a DNA template containing a T7 promoter on both ends, the enzyme T7 RNA polymerase, and ribonucleoside triphosphates (rNTP). Then, dsRNA was purified by precipitation with lithium chloride. This method removes most of the protein, NTP, and DNA template. RNA purity was visualized by gel electrophoresis. Next, dsRNA concentration was measured using a Nanodrop and the dsRNA shipped at room temperature dissolved in water. We trialled 22 potentially-lethal dsRNA sequences as listed in S1 Table, either alone, concatenated or combined in a mix or 'cocktail' (see below). The rationale for the design of the dsRNAs followed criteria outlined in Knorr et al. [27] with slight modifications. In general, dsRNA sequences were designed to be between 200–500 bp (aiming for as close to 500 bp as possible); targeting a gene region no closer than 70 bp

from the ATG or stop codon; a GC% between 40–60 where possible; and the highest number of potential siRNA sequences as determined using http://sidirect2.rnai.jp. The concatenated sequences were much bigger (long A dsRNA = 1306 bp; long B dsRNA = 1472 bp and long C dsRNA = 993 bp; S2 Table) and did not follow the same design criteria. Instead these sequences were based on the dsRNA developed to target the pest *Varroa destructor* [30, 33].

## Different methods of dsRNA delivery to paper wasps

**Injecting dsRNA into adult forager wasps.**  Sixty-five foragers were collected at random from the different captive colonies and placed in individual, transparent jars (5 x 5 x 6 cm); provided with sugar candy and drinking water for 48 h prior to the experiment. We injected 12 different dsRNAs directly into the wasp abdomen. Treatment consisted of injecting five wasps per gene target: FACT complex subunit spt16 (DRE4); RNA-binding protein fusilli (FUSILLI); charged multivesicular body protein 4b (DVSNF7); tubulin alpha-1 chain (α-TUBULIN); kinesin-like protein KIF11-A (KLP61F-1); Anthonomus grandis chitin synthase II (AGRA); CACTIN; DNA-directed RNA polymerase II subunit RPB2 (RPII140); ras opposite (ROP); NF-kappa-B inhibitor cactus (CACT); putative U5 small nuclear ribonucleoprotein 200 kDa helicase (BRR2); and CWC22 homolog (NCM). The five wasps in the control group were injected with the same volume of water as a control. For the injection assays adult wasps were sedated with $CO_2$ and restrained with soft-tipped forceps by one researcher while the other researcher delivered the injection. Wasps were injected between the V or VI abdominal tergites (Fig 1). Each wasp was injected with 1 μL of dsRNA solution at a concentration of 5 μg/μL (equivalent to 5 μg per wasp) using a 900 series, model 95 Hamilton syringe fitted with a 26-gauge needle, point style 2 (Hamilton Company, Nevada, USA). Following injection, wasps were returned to their individual plastic jars, left there for 48 h, and subsequently frozen at -80˚C. If a wasp died before 48 h, it was frozen immediately after being found dead and its death recorded.

**Orally feeding dsRNA to adult forager wasps.**  *Assay 1. Comparing survival of adult wasps when feeding on a 'cocktail' of unprotected dsRNA versus sugar water control*. We performed an experiment to determine if feeding adult foragers a cocktail of four dsRNA gene targets, combined in one solution, could affect wasp survival. We first immobilised a forager wasp inside a 1000 μL pipette tip with the end cut and provided them with a random dsRNA to drink from the end of a pipette tip, to ensure they will accept the liquid, which was accepted and swallowed swiftly (Fig 2). In this trial, the control group consisted of 12 wasps, individually kept in their own jar (5 x 5 x 6 cm) and fed 50 μL of sugar water once daily for 10 days, while the treatment group consisted of 12 wasps fed 50 μL of a dsRNA cocktail at a concentration of 0.5 μg/μL once daily for 10 days. We pipetted the drop of liquid into the floor of the cage, on top of a square of parafilm (Bemis Company, Wisconsin, USA). The cocktail consisted of four dsRNA genes combined in one solution (α-TUBULIN, DVSNF7, DRE4, and FUSILLI). Survival was recorded for the 10 days and wasps were then frozen alive at -80˚C.

*Assay 2. Assessing gene knockdown following single and concatenated dsRNA feeding*. In this larger scale experiment we fed individual wasps with single target dsRNA genes and concatenated dsRNA genes to assess mortality. Each group consisted of seven forager wasps individually kept in its own container (5 x 5 x 6 cm) and randomly assigned to one of seven different treatments: sugar water (control group); calmodulin-like (CaM) dsRNA; similar to 26S proteasome non-ATPase regulatory subunit 6 (RPN7) dsRNA; fork head (FKH) dsRNA; long A concatenated dsRNA; long B concatenated dsRNA; and long C concatenated dsRNA, each targeting multiple genes (see below). Each treatment corresponded to a solution of 1 μg/μL of dsRNA. Each wasp was fed 50 μL per day of the corresponding treatment for 7 days. The

**Fig 1. A photograph showing how the dsRNA injections were delivered into the paper wasp's abdomen.** Photograph credit: Phil Lester.

liquid was pipetted on top of the sugar candy placed on the container floor. The long A dsRNA was 1,306 bp and targeted V-type proton ATPase 116 kDa subunit a-like (VATPase) and DNA-directed RNA polymerase III subunit RPC2-like (RPC2); long B dsRNA was 1,472 bp and targeted α-TUBULIN, Apoptosis inhibitor 5-like (APOPIN) and DNA-directed RNA polymerase I subunit RPA1-like (RPA1) dsRNA; and long C dsRNA was 993 bp and targeted putative inhibitor of apoptosis (INAPOP) and sodium/potassium-transporting ATPase subunit beta-2-like (NaKATPase) dsRNA. Following the 7 days, wasps were frozen alive at -80˚C. We compared gene expression via RT-qPCR (see below). For the analyses of the qPCR results,

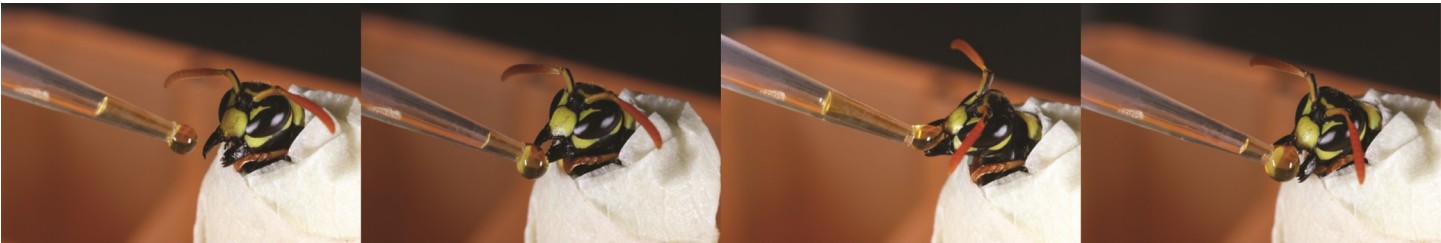

**Fig 2. Feeding an immobilised *P. dominula* forager with a dsRNA solution to ensure that the wasps consumed the dsRNA solution provided in the cages, previous to the trials.**

we compared expression of each dsRNA targeted-gene relative to the expression of the same gene in wasps fed just sugar water. For the different long concatenated dsRNAs that targeted multiple genes, we determined gene expression independently for each of the targets.

*Assay 3. Nest-level dsRNA feeding treatment.* This trial consisted of providing sugar sources with either dsRNA solutions or sugar water to whole colonies. Individual nests were placed in large, plastic, transparent cages (22 x 22 x 22 cm, Fig 3) and left to acclimate for a day. Each cage only held one nest. Food and water were provided *ad libitum*. Trials consisted of three nests per treatment group, each nest was provided once daily with 750 μL of the corresponding dsRNA at a concentration of 0.5 μg/μL for 7 days. The treatment consisted of providing (I) a sugar water solution on top of the fondant candy as control group, (II) GFP on top of the fondant candy, (III) a cocktail that consisted of the three long A, B, C dsRNAs used in assay 2 combined, and (IV) a 'mega-cocktail' of dsRNAs that targeted 16 genes combined in one solution (RpII140, DRE4, NCM, ROP, RPB7, BRR2, Klp61F-1, α-TUBULIN, CACT, CACTIN, FUSILLI, DVSNF7, AGRA, RPN7, FKH and CaM). Any wasp mortality was recorded daily. Following the 7 days, wasps were frozen alive at -80°C until RNA extraction. Gene expression of wasps from each of the three treatments (II, III, IV) was compared to the sugar water control (I) via RT-qPCR. For the dsRNA cocktails, we selected one gene target from each mixture (VATPase for long A dsRNA, RPA1 for long B dsRNA, and NaKTPase for long C dsRNA) to examine in the qPCR analyses. For the 'mega-cocktail', we determined gene expression via RT-qPCR of DRE4 and CaM relative to the sugar water control.

*Assay 4. Comparing gene knockdown of naked dsRNA versus CQD-protected dsRNA.* This experiment consisted of providing dsRNA targeting the same gene in two preparations, naked dsRNA and nanoparticle-protected dsRNA. We selected four gene targets: α-TUBULIN; DVSNF7; DRE4; and FUSILLI, based on the results of the injection trials. dsRNA was protected by conjugation to Carbon Quantum Dots (CQDs) [34] (see below). We collected wasps from the different captive nests, and kept each individual wasp in its own plastic, transparent jar (5 x 5 x 6 cm) provided only with drinking water for 24 h prior to the experiment. Each trial consisted of 12 wasps per treatment group, with four treatments per gene for a total of 48

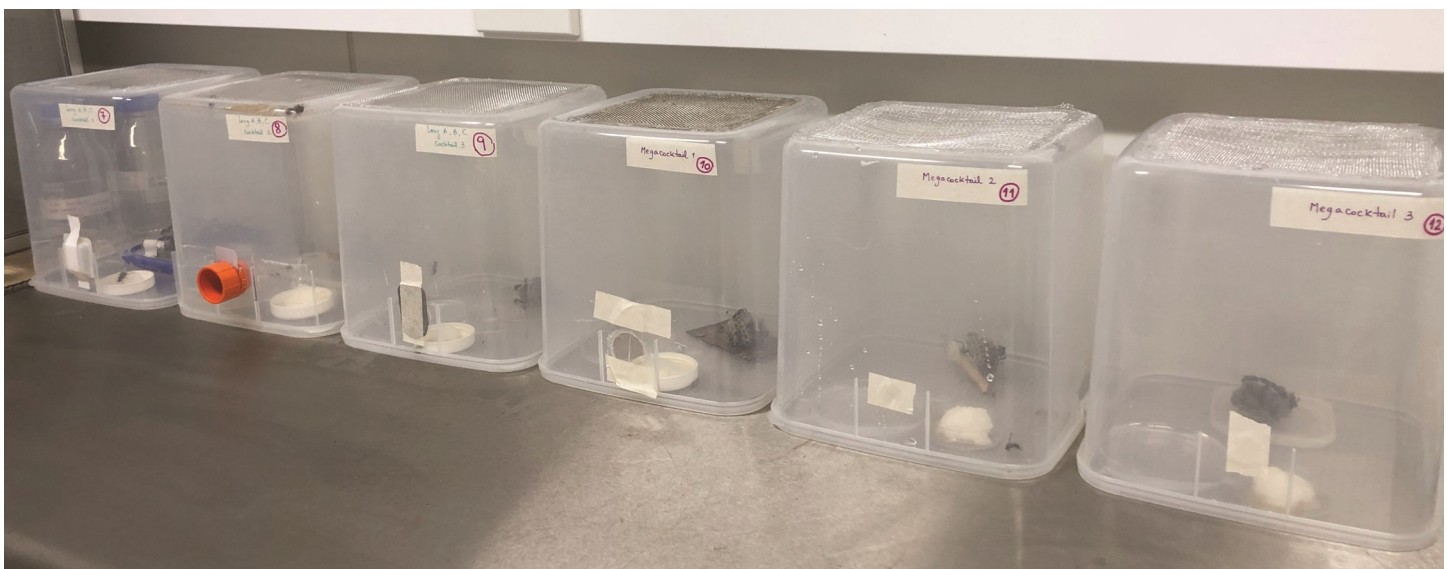

**Fig 3. A photograph showing the experimental set up when providing whole colony access to the dsRNA solution.** Each cage held one nest, with foragers able to return to their nests and tend for the larvae. Photograph credit: Mariana Bulgarella.

wasps per gene target per concentration (0 μg/μL of dsRNA = sugar water control; 0.5 μg/μL of dsRNA, 1 μg/μL of dsRNA, and 5 μg/μL of dsRNA). Each container had a fondant candy ball into which we delivered 50 μL of the target dsRNA of the corresponding concentration once daily for 10 consecutive days, which the wasps drank readily. Water was provided *ad libitum* and survival recorded daily for the 10 days of the experiment. If an individual was found dead, it was recorded and removed from the experiment. On day 10, all surviving wasps were frozen at -80˚C. To examine dsRNA effects, the expression of each of the four target genes was determined by RT-qPCR relative to the sugar water controls.

For the preparation of dsRNA-CQD conjugates, a stock CQD solution was prepared following the approach of Kaur et al. [9]. Nine mL of poly(ethylene glycol) (Sigma Aldrich, Missouri, USA) was combined with 3 mL nuclease-free $H_2O$ (Invitrogen, Massachusetts, USA), and 2 mL of 50 mg/mL polyethylenimine solution (branched, average Mw ~25,000 by LS, average Mn ~10,000 by GPC, Sigma Aldrich, Missouri, USA). The 14 mL mixture was heated in the microwave in short bursts for a total of 3 min and then allowed to cool. Volumes of this stock CQD solution were diluted 1:10, 1:100, and 1:200 prior to conjugation to the three selected concentrations of dsRNA. These dilutions were determined by gel retardation assays in which the diluted CQD bound all applied dsRNA. CQD-dsRNA conjugates at three different RNA concentrations were prepared as follows. For the 0.5 μg/μl treatments 3.5 mg of the dsRNA solutions were made to 7 mL with 50 mM sodium sulfate (Sigma Aldrich, Missouri, USA) and then added to 7 mL of 1:200 diluted CQD solution. For the 1 μg/μL treatments 7 mg of the dsRNA solutions were made to 7 mL with sodium sulfate and added to 7 mL of 1:100 diluted CQD solution. For the 5 μg/μL treatments 35 mg of the dsRNA solutions were made to 7 mL with sodium sulfate and added to 7 mL of 1:10 diluted CQD solution. The mixtures quickly generated a precipitate in the 15 mL tubes and were distributed across seven 2 mL tubes which were centrifuged for 10 min at 7,000 g and the supernatant discarded. The pellets made of CQD-dsRNA conjugates were resuspended by pipetting and gentle sonication in 1 mL of 50% sugar water per tube and then combined for a total of 7 mL per treatment.

*Assay 5. Comparing gene knockdown of unprotected dsRNA versus lipofectamine-protected dsRNA.* Adult foragers were collected from the different captive nests and put in individual plastic, transparent jar (5 x 5 x 6 cm) provided with only drinking water for 24 h prior to the experiment. For the feeding assay, 100 μL of dsRNA at a concentration of 0.5 μg/μL was pipetted on top of the candy fondant placed on the jar floor twice daily (at 9 am and 4 pm) for 4 days. Treatments consisted of feeding 10 individual wasps in each treatment group (termed naked DRE4 dsRNA, naked RPII140 dsRNA, naked GFP dsRNA, lipofectamnine-DRE4 dsRNA, lipofectamnine-RPII140 dsRNA and lipofectamine-GFP dsRNA). Lipofectamine is a commonly used transfection reagent that forms liposomes (lipid nanoparticles) when mixed with nucleic acids. dsRNA liposomes were formed by mixing 500 μL Lipofectamine 3000 Transfection Reagent (Invitrogen/Thermo Fisher Scientific, Massachusetts, USA) with 500 μg of the target dsRNA, and made to 1 mL final volume with sugar water.

Following the 4 days of dsRNA feeding, wasps were frozen alive at -80˚C. If a wasp died before then, it was recorded and removed from the experiment. qPCR was used as described below to determine relative gene expression, using GFP as the control group.

**RNA extraction, reverse transcription and quantitative PCR.** To determine gene expression changes, we extracted RNA using the whole body from individual wasps via a C-TAB/chloroform-style extraction protocol that simultaneously extracts both RNA and DNA. Each whole wasp sample was homogenized in a microcentrifuge tube containing 1 mL of GENEzol plant DNA reagent (Geneaid Biotech, Taiwan), 5 μL of β-mercaptoethanol (Sigma Aldrich, Missouri, USA) and two 5 mm stainless steel beads in a Precellys Evolution homogenizer (Bertin Instruments, France). Next, RNA was isolated using a 24:1 chloroform–isoamyl

alcohol mixture (BioUltra, Sigma Aldrich, Missouri, USA) followed by an isopropanol precipitation step (BioReagent, Sigma Aldrich, Missouri, USA), and a 70% ethanol purification step (VWR Chemicals, UK). Lastly, RNA was eluted in 100 μL of nuclease-free water (Ambion, Life Technologies, Massachusetts, USA). The RNA was quantified using a NanoPhotometer NP80 (Implen, Germany), and 1 μg of RNA was reversed transcribed (RT) with Quanta qScript XLT cDNA SuperMix (Quantabio, Massachusetts, USA) in 20 μL reactions following the manufacturer's instructions.

Gene expression was measured by quantitative polymerase chain reaction (qPCR). We designed gene-specific primer pairs for each of the dsRNA-targeted genes that bound to regions outside of the dsRNA (S2 Table). Two genes were selected as internal reference genes chosen based on their stability in previous hymenopteran research [35]: NADH dehydrogenase [ubiquinone] 1 alpha subcomplex subunit 8 (NDUFA8, the orthologue of a component of complex one of the mitochondrial electron transport chain) and 26S proteasome non-ATPase regulatory subunit 4 (PROS54). Sample cDNA was diluted to 2 ng/μL with nuclease-free water and 8 μL combined with a 12 μL mix containing PowerUp SYBR Green Master Mix (Applied Biosystems/ThermoFisher Scientific, Massachusetts, USA), and forward and reverse primers (final concentrations of 300 nM). Quantitative PCR was conducted in 96-well plates on a QuantStudio 7 Flex Real-Time PCR platform (Applied Biosystems/ThermoFisher Scientific, Massachusetts, USA). The following fast cycling conditions were used: 50˚C, 2 min; 95˚C, 2 min; 40 cycles of 95˚C, 1 s, 60˚C, 30 s. Fluorescence was measured at the 60˚C step and quantification cycle (Cq) values averaged from two technical replicates were used to calculate gene expression levels relative to the reference genes NDUFA8 and PROS54. Gene expression was calculated by normalising the raw target gene mRNA expression values to the average of the reference gene mRNA expression values using the equation $(2^{\wedge}(-Cq$ target gene$))$/average of $(2^{\wedge}(-Cq$ NDUFA8$))$ and $(2^{\wedge}(-Cq$ PROS54$))$ [36, 37]. Relative gene expression was calculated using the average of the experimental samples divided by the average of control samples and is reported along with the standard error of the mean (SE).

### Data analyses

All analyses were performed within the R statistical environment [38]. Due to non-normality of qPCR data, non-parametric Kruskal-Wallis ANOVA tests were used to compare differences in gene expression [23] followed by Dunn post hoc tests where appropriate with *p* values adjusted for multiple comparisons with the Benjamini-Hochberg method. Graphs from the qPCR data were generated using *ggplot2* in R [39]. We present mean ± 1 standard error (SE) of the mean throughout the manuscript.

## Results

### Different methods of dsRNA delivery to paper wasps

**Injecting dsRNA into adult forager wasps.** In this experiment we injected 12 different dsRNAs, known to be lethal in other insects, directly into the abdomen of wasps. Overall, five of the targeted genes presented decreased expression 48 h after dsRNA injection: DRE4, FUSILLI, DVSNF7, α-TUBULIN and KLP61F-1 (Fig 4). Only DRE4 and FUSILLI, however, showed a significant reduction of expression (Kruskal-Wallis $\chi^2$ = 3.94, *p* = 0.04), and could potentially be promising candidates for lethal gene silencing. We found varying levels of gene expression response to the dsRNA injections (Fig 4). Contrary to our expectations, some of the dsRNA-gene targets showed increased expression following injection compared to wasps that had been injected water.

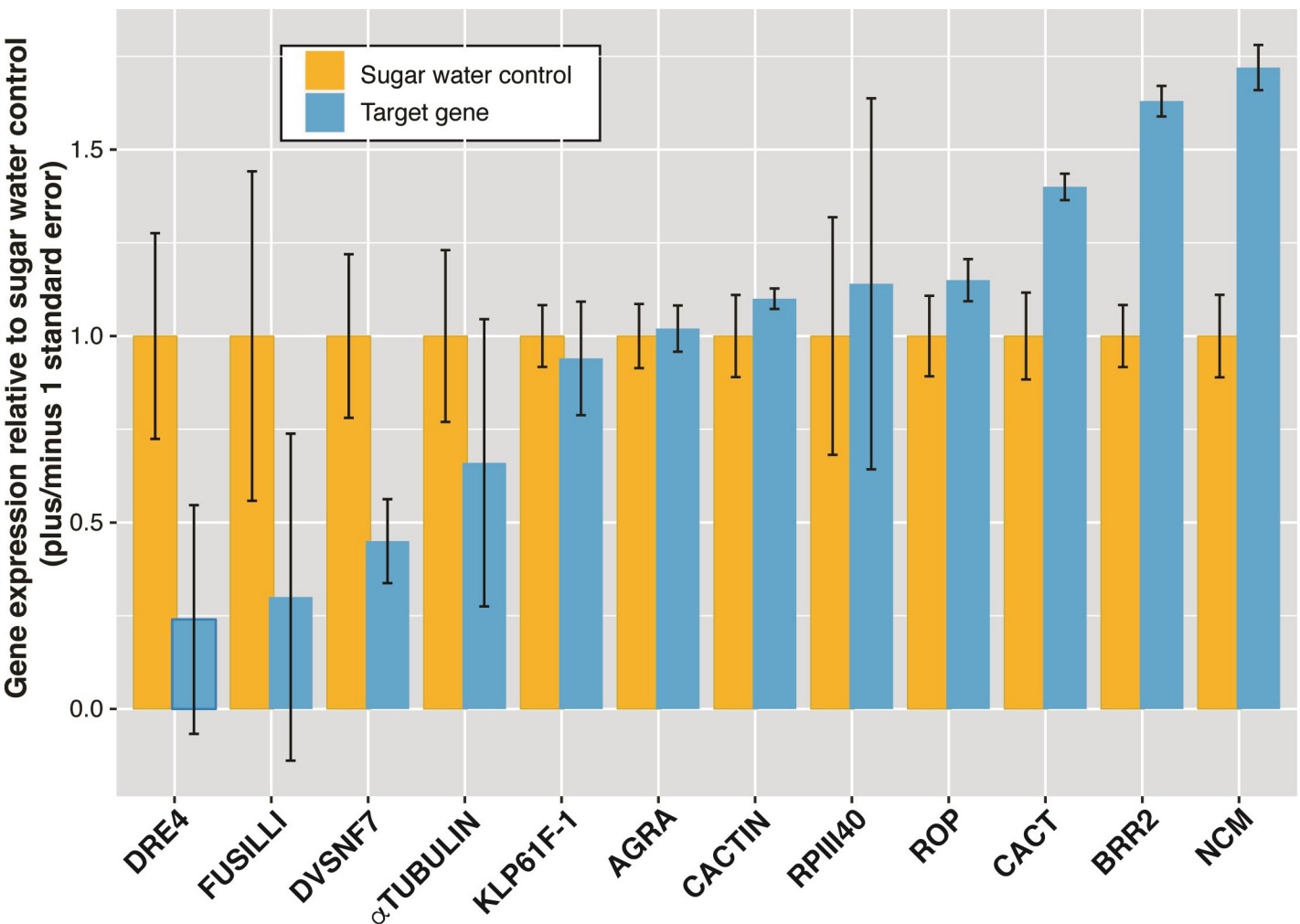

**Fig 4. Relative gene expression for *P. dominula* wasps 48 h following dsRNA injections.** Twelve different gene targets were injected into the wasps. Each bar represents the mean expression value pooled for five wasps and the SE compared to five control wasps injected with water.

Wasp mortality was monitored following the dsRNA injections for 48 hours as previous studies suggest that the silencing effect of dsRNA might be transient [23, 40]. Very little mortality was observed in any treatment. Out of 60 wasps injected dsRNA, five wasps died 24 h after injections (8.3%), and four wasps died 48 h after injections (6.6%). Out of the five wasps in the control group, only one died 24 h following injection. Our goal in these trials with only a limited number of wasps was to identify gene targets that would cause significant mortality in wasps, and then perform trials with larger sample sizes to better define mortality. However, because only trivial levels of mortality were observed for all gene targets we did not pursue larger trials.

**Orally feeding dsRNA to adult forager wasps.** *Assay 1. Comparing survival of adult wasps when feeding on a 'cocktail' of unprotected dsRNA versus a sugar water control.* This experiment examined the mortality of adult foragers fed a cocktail of four dsRNA gene targets (α-TUBULIN, DVSNF7, DRE4, and FUSILLI), combined in one solution, could influence wasp survival. Wasps were fed the dsRNA cocktail every day for 10 days. The wasps were observed to consume the sugar water and dsRNA presented to them. The results of this experiment showed that this dsRNA cocktail had no apparent effect on mortality for adult

*P. dominula* kept in captivity. Out of 24 wasps, one wasp in the control group died on day 4, and one wasp in the treatment group died on day 1. All remaining 22 wasps were alive after 10 days of feeding on a dsRNA cocktail or a sugar water solution without any dsRNA.

*Assay 2. Assessing gene knockdown following single and concatenated dsRNA feeding.* In this trial, our aim was to determine gene expression following ingestion of dsRNA targeting a single dsRNA gene or concatenated set of dsRNAs targeting multiple genes. The relative gene expression results of feeding adult female wasps on a single dsRNA or concatenated dsRNAs show no differences in gene expression between any of the six treatments and the respective control groups (Kruskal-Wallis $p > 0.05$, Fig 5 and S3 Table).

One benefit of this experiment was to confirm that wasps were feeding on the dsRNA. For the calmodulin (CaM) gene target, the qPCR primers unavoidably overlapped the dsRNA

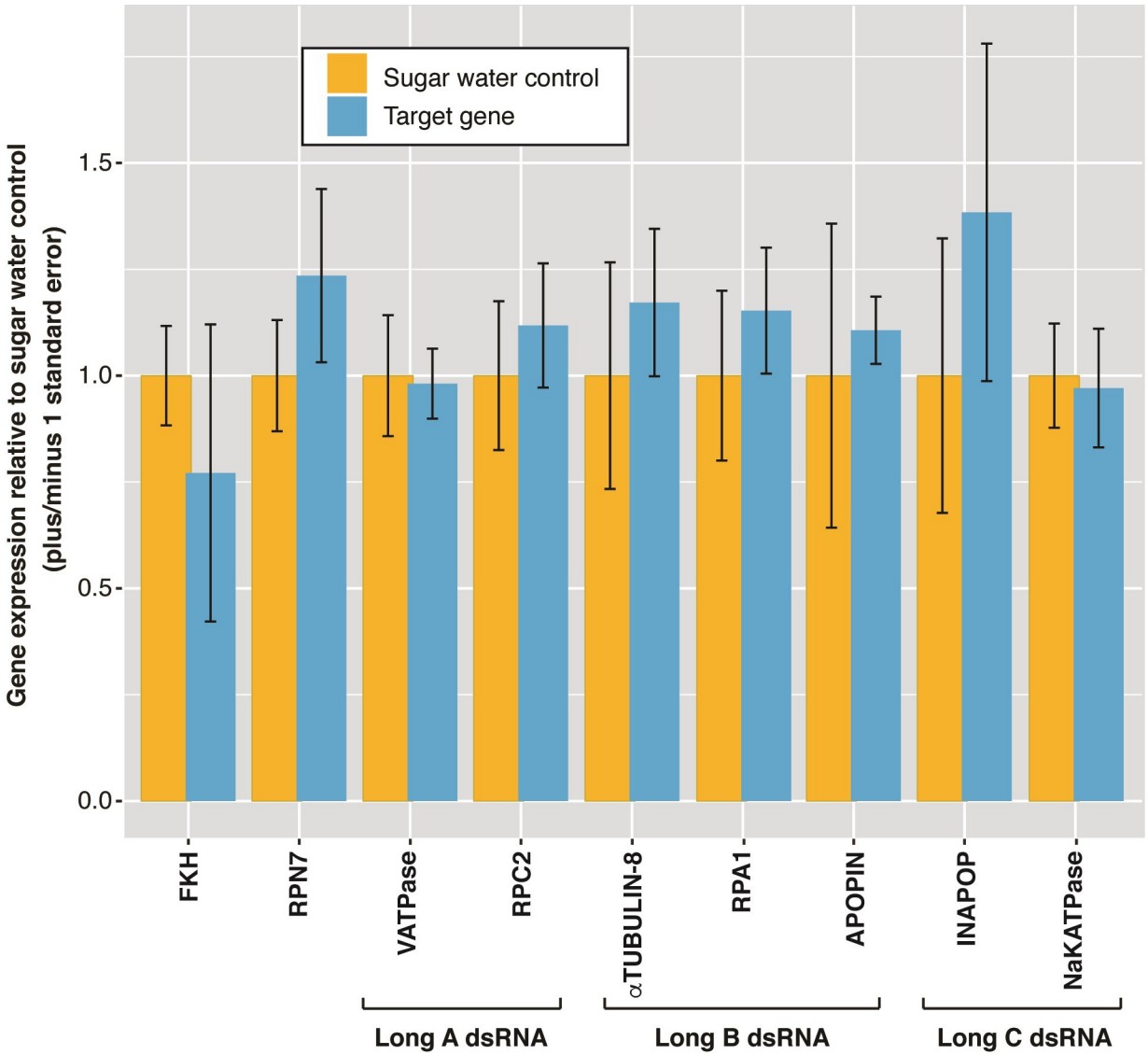

**Fig 5. Relative gene expression for single gene dsRNA targets fed alone (FKH and RPN7) or in concatenated form to forager *P. dominula* when compared to the sugar water controls.** Each bar represents the pooled mean ± SE for three wasps, compared to three control wasps. Note that CaM is not shown in this graph but in S1 Fig.

region. The RT-qPCR clearly demonstrated CaM dsRNA consumption as shown in the apparent massive increase in CaM gene 'expression' due to the qPCR primers binding to the reverse-transcribed dsRNA, rather than indicating an increase in gene expression (S1 Fig). This result also indicated that the lack of gene knockdown in other experiments is not due to the wasps not feeding on the dsRNA.

No mortality was recorded for wasps fed long A, long B, and long C concatenated dsRNAs, CaM dsRNA, RPN7 dsRNA or the sugar water control. One wasp fed FKH dsRNA died on day 5.

*Assay 3. Nest-level dsRNA feeding treatment.* In this trial, we offered a solution of dsRNA or sugar water to whole colonies. Forager wasps drank the solutions readily, as observed with individual wasps previously. We also observed the foragers then feed the larvae in their nests with the solution. No other liquid source of food was available but wax moth were provided for feeding larvae. In this experiment we meant to mimic a more natural environment in which the wasps are able to access their nests of origin and tend for their young. The dsRNA was delivered daily for 7 days. We found no significant differences in relative gene expression for wasps fed the dsRNA cocktails when compared to control wasps fed sugar water (Kruskal-Wallis $p > 0.05$, Fig 6). To determine gene expression, we chose one gene in each cocktail mix and compared it to the expression of GFP and to that of sugar water as controls (S3 Table).

No mortality was recorded for larvae and pupae on the nests and extremely low for foragers, during the 7 day trial duration. One wasp died on day 1 in the long A, B, C cocktail combined treatment, one wasp died on day 1 on the megacocktail dsRNA treatment and one wasp died on day 1 on the GFP dsRNA group.

*Assay 4. Comparing gene knockdown of unprotected dsRNA versus CQD-protected dsRNA.* As previous dsRNA ingestion trails showed only minimal gene knockdown and no mortality, we decided to provide dsRNA for a particular gene target in two presentations: naked dsRNA and nanoparticle-protected dsRNA. The resulting patterns of relative gene expression was not consistent among genes or concentrations tested. FUSILLI dsRNA did not cause decreased gene expression when compared to the sugar water control, independent of the dsRNA being unprotected or protected, and at any of the concentrations tested (Kruskal-Wallis $\chi^2 = 3.70$, df = 6, $p = 0.72$). DVSNF7 showed a significant decrease in relative expression when compared to the sugar water control (Kruskal-Wallis $\chi^2 = 18.15$, df = 6, $p = 0.005$). When multiple comparisons were performed, we found significant differences between the unprotected DVSNF7 dsRNA and the CQD-protected DVSNF7 dsRNA at a concentration of 1 μg/μL (Dunn test $Z = 2.65$, $p_{adjusted} = 0.05$), with relative expression decreasing significantly for the unprotected DVSNF7 dsRNA at 1 μg/μL. For α-TUBULIN, the largest knockdown occurred with 1 μg/μL dsRNA treatment (Fig 7) but the differences were not statistically significant (Kruskal-Wallis $\chi^2 = 5.82$, df = 6, $p = 0.44$). For DRE4, we only fed the wasps with unprotected-DRE4 dsRNA, and observed a slight, not statistically significant, decrease in gene expression at all three concentrations tested (Kruskal-Wallis $\chi^2 = 2.13$, df = 3, $p = 0.55$, Fig 7).

Similarly to the injection results and previous dsRNA ingestion experiments, the survival data shows that the wasps did not die in substantial numbers, nor consistently, in response to dsRNA feeding. No substantive increase in mortality was observed following feeding with the unprotected dsRNA form or in the treatment with CQD-protected dsRNA conjugates (S2 Fig and S3 Table).

*Assay 5. Comparing gene knockdown of naked dsRNA versus lipofectamine-protected dsRNA.* In this assay, we used lipofectamine to protect dsRNA from any digestive degradation after wasp feeding, and this assay presented the most promising results in terms of gene silencing. When DRE4 dsRNA was protected by lipofectamine encapsulation, DRE4 gene expression

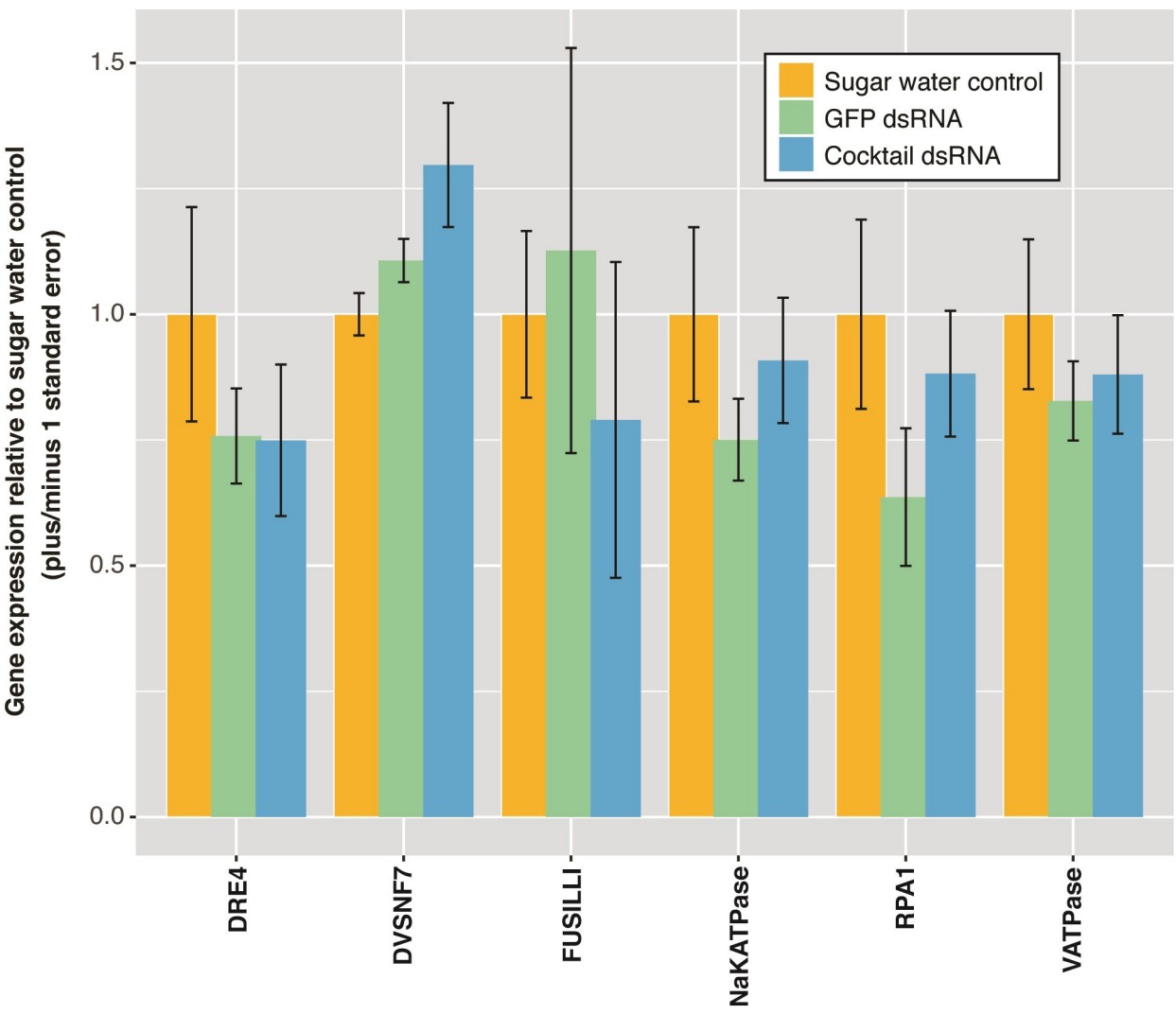

**Fig 6. Relative gene expression of *P. dominula* foragers provided dsRNA while able to access to and tend for their nests of origin, in a set up mimicking more natural conditions.** Bars correspond to the pooled mean ± SE for six wasps (randomly selected) in each dsRNA treatment, compared to three wasps fed GFP dsRNA and three control wasps fed sugar water.

significantly decreased compared to the GFP dsRNA control (Kruskal-Wallis $\chi^2$ = 20.06, df = 3, $p$ < 0.001, Fig 8). Relative expression for DRE4 following treatment with GFP dsRNA encapsulated with lipofectamine was significantly lower than unprotected GFP dsRNA treatment (Dunn test $Z$ = -2.78, $p_{adjusted}$ = 0.01). Expression of DRE4 following treatment with DRE4 dsRNA encapsulated with lipofectamine was significantly reduced compared with expression following unprotected GFP dsRNA treatment (Dunn test $Z$ = 3.88, $p_{adjusted}$ < 0.001). GFP encapsulated with lipofectamine resulted in significantly lower DRE4 expression than feeding wasps with unprotected DRE4 dsRNA (Dunn test $Z$ = -2.22, $p_{adjusted}$ = 0.03). Finally, when DRE4 dsRNA was protected by lipofectamine encapsulation, DRE4 gene expression decreased significantly compared to the relative expression following unprotected DRE4 dsRNA feeding (Dunn test $Z$ = -3.34, $p_{adjusted}$ = 0.002).

The relative expression of RPII140 significantly increased following treatment with respect to the GFP dsRNA controls, independently of encapsulation status (Kruskal-Wallis $\chi^2$ = 19.38,

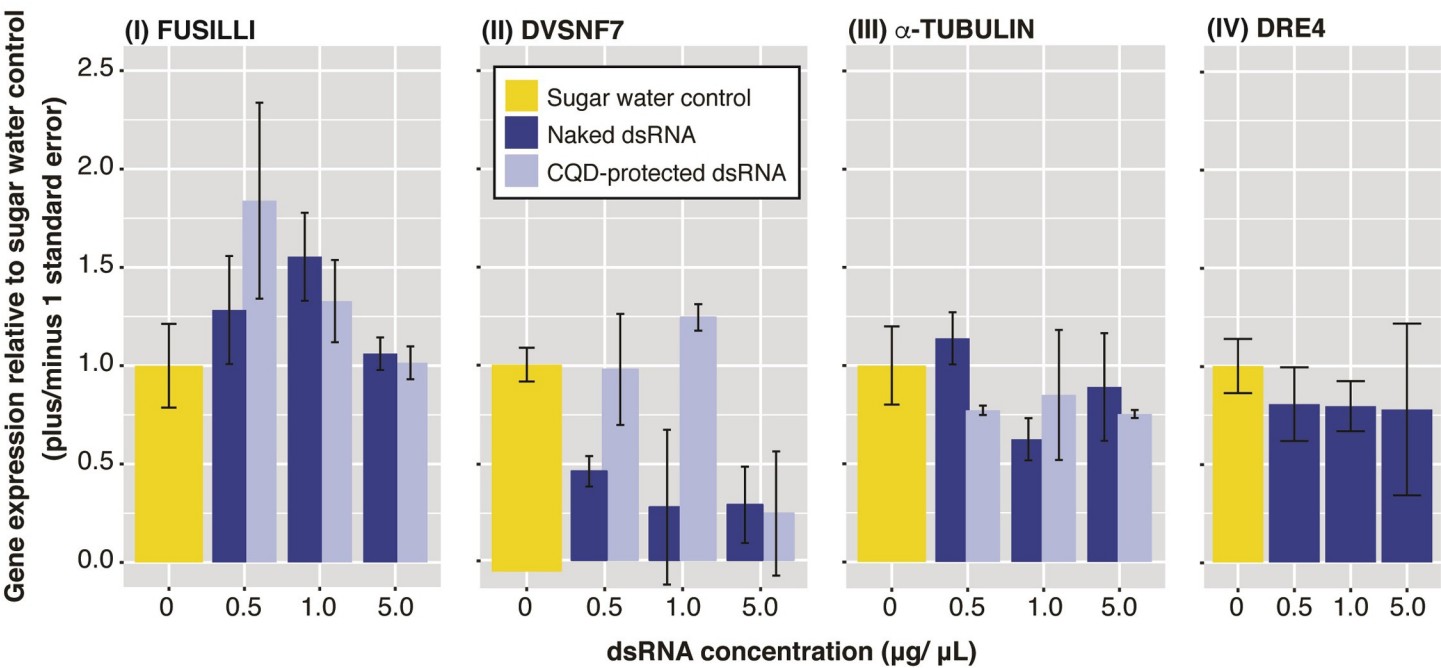

**Fig 7. Relative gene expression for *P. dominula* wasps fed four dsRNAs gene targets of different concentrations with and without CQD protection.** Each bar corresponds to the pooled mean ± SE for three wasps, compared to three wasps in a control treatment that were fed only sugar water (with no dsRNA).

df = 3, $p < 0.001$, Fig 8 and S3 Table). Relative expression of RPII140 following lipofectamine-encapsulated GFP dsRNA increased significantly when compared to the unprotected GFP dsRNA (Dunn test $Z = 3.24$, $p_{adjusted} = 0.003$), the opposite pattern shown by DRE4 dsRNA. Expression of RPII140 following lipofectamine-encapsulated RPII140 dsRNA treatment increased significantly compared to the unprotected GFP dsRNA control (Dunn test $Z = -3.55$, $p_{adjusted} < 0.001$). Lipofectamine-encapsulated GFP dsRNA treatment resulted in significantly higher expression of RPII140 than unprotected RPII140 dsRNA treatment (Dunn test $Z = 2.54$, $p_{adjusted} = 0.01$). Lipofectamine-protected RPII140 dsRNA increased RPII140 expression significantly compared to treatment with unprotected RPII140 dsRNA (Dunn test $Z = 2.85$, $p_{adjusted} = 0.008$), the opposite to what we observed with DRE4 dsRNA (Fig 8).

We only recorded one death in this assay, when a wasp fed DRE4 dsRNA died on day 3 of the experiment.

## Discussion

The goal of our study was to develop a species-specific method for the control of adult paper wasps, by targeting genes that when silenced would lead to mortality. To our knowledge, this study constitutes the first attempt to use dsRNA for pest management of invasive social wasps. The provision of dsRNA to adult wasps, either orally or injected suggested some level of gene silencing but no substantial mortality was achieved. Two gene targets were successfully knocked down in our assays, FACT complex subunit SPT16 (DRE4) and charged multivesicular body protein 4b (DVSNF7). The DRE4 gene target is part of the FACT (facilitates chromatin transactions) complex and acts as a general chromatin regulator, recognizing nucleosomes [27, 41]. Based on a list of lethal RNAi targets for the red flour beetle, *Tribolium castaneum*, Knorr et al. [27] identified the ortholog sequences of DRE4 in the pollen beetle, *Meligethes aeneus*, and the western corn rootworm beetle, *Diabrotica virgifera virgifera*. When dsRNAs

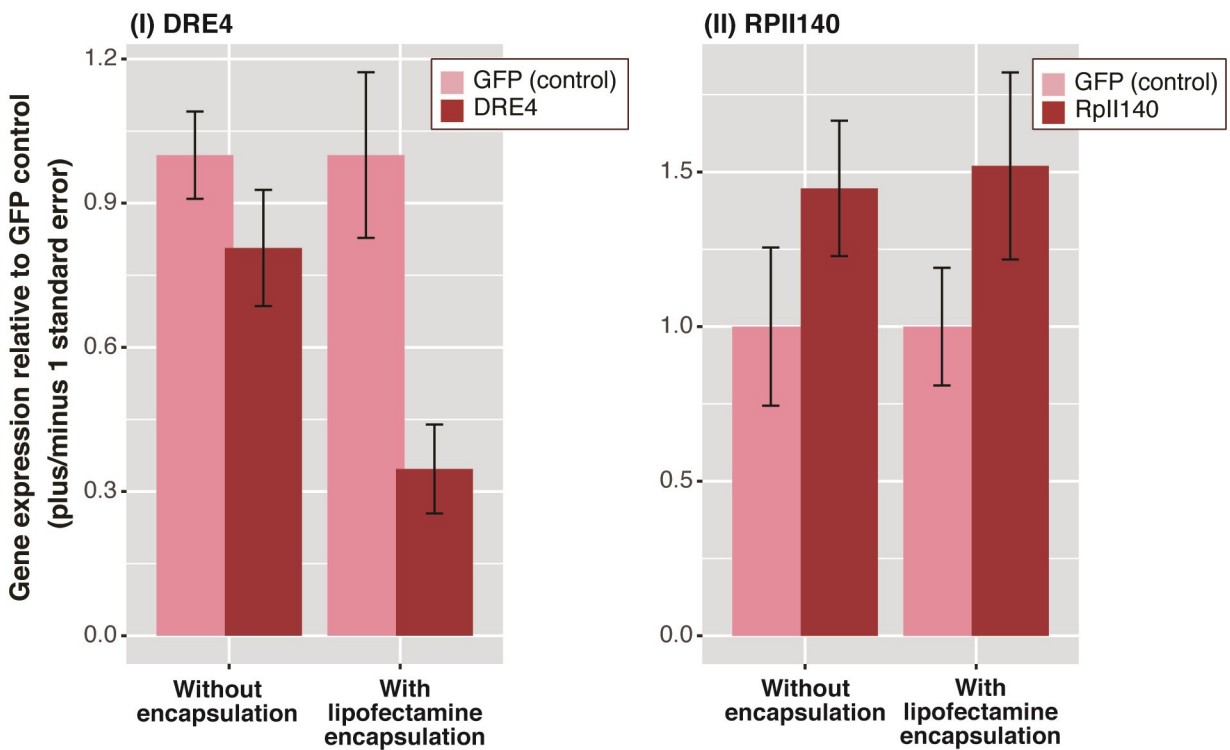

**Fig 8. Relative gene expression for *P. dominula* foragers that fed on two gene targets: (I) DRE4 and (II) RPII140 dsRNA, with and without lipofectamine encapsulation.** For the encapsulated dsRNA, each bar corresponds to the pooled mean ± SE of five wasps, compared to five GFP control wasps. For the unprotected dsRNA, each bar corresponds to the pooled mean ± SE for nine wasps, compared to the control treatment of nine wasps fed GFP dsRNA.

were injected or fed to adult beetles, gene knockdown and high mortality was achieved, and the authors highlighted this gene target as a promising candidate for insect pest control [27]. This is the same approach we followed, and although gene knockdown was achieved, we did not observe adult wasp mortality. The second gene target successfully silenced was the DVSNF7 (*D. virgifera* SNF7) protein, which functions as a part of the ESCRT (Endosomal Sorting Complex Required for Transport)-III complex, a crucial pathway for transmembrane protein sorting [42, 43]. Upon consumption, the plant-produced DVSNF7 dsRNA is recognized by the western corn rootworm's RNAi machinery, which results in down-regulation of the DVSNF7 gene and ultimately killing the beetle [26, 43]. DVSNF7 dsRNA-expressing maize (*Zea mays*) targeting the western corn rootworm beetle was the first insecticidal dsRNA-expressing plant registered by the US Environmental Protection Agency [44].

There have been two previous studies of gene silencing in paper wasps, with both having the objective of determining gene function [22, 23]. In one of the studies when dsRNA was fed to fifth instar paper wasp larvae, gene expression was reduced without achieving observable phenotypic effects [22]. These are similar results to the ones reported in our study. In the second paper wasp study, dsRNA was fed to second and third instar paper wasp larvae, and one of two target genes was knockdown but only when gene expression was measured one day after dsRNA feeding. No differences in gene expression were found when larvae fed dsRNA were left to pupate before collection. No elevated mortality was recorded either but a small decrease in lipid stores following one dsRNA treatment was observed in the shorter post-treatment timeframe. The authors point to the highly transient nature of knockdown effects, which may even be limited to 24 hours or less following the dsRNA treatment [23, 27]. In our study,

we varied the times post-dsRNA treatment in the different assays in order to account for the potential different timeframes optimal to observe gene silencing and/or mortality effects. Further, these previous trials with feeding paper wasps dsRNA have targeted larvae, as these earlier trials aimed to determine the function of specific genes in the development of reproductive castes. Our main objective in this study was to develop a species-specific control programme for widespread use in the field. The developmental stage that we targeted was the adult forager wasps which would collect, bring back and distribute the dsRNA in their colonies. It would be unrealistic to feed and monitor individual larva at such large scale, which is why in our study we focused on adult survival.

Explanations for the lack of consistent knockdown or mortality in our assays include, but are not limited to, possible dose response, a dilution effect in the trials with entire colonies when dsRNA may have been shared via trophallaxis in the colonies, and dsRNA degradation by gut nucleases within the adult wasps.

Different concentrations of dsRNA fed to insects have different efficiencies in initiating a RNAi response [45]. We tested four different concentrations of dsRNA, and achieved successful knockdown at concentrations of 0.5 μg/μL (assay 5, lipofectamine-protected DRE4 dsRNA, a total of 200 μg of dsRNA per wasp) and 1 μg/μL (assay 4, unprotected DVSNF7 dsRNA, a total of 500 μg of dsRNA per wasp over the course of the trial). In our experiments, it would seem that feeding the wasps a more concentrated dose of dsRNA (5 μg/μL) did not increased gene knockdown. Turner et al. [45] argues that the midgut environment of different insects may require different concentrations of dsRNA to trigger silencing. Therefore, it is important to test various concentrations of dsRNA and optimise the one that works for your study species [45]. Hunt et al. [22] reported successful gene knockdown in *Polistes metricus* larvae by using doses of dsRNA from honey bees and termites studies (20 μg of dsRNA given twice a day for 2 days, for a total of 80 μg of dsRNA per larvae).

Even though we effectively knocked down genes which have been shown to result in lethal effects in other insects, their reduced expression did not cause mortality in wasps. When a cocktail of dsRNAs (which included DRE4) was provided to wasps with access to their nests, we observed a slight but non-statistically significant reduction of expression of some genes relative to the sugar water controls. One possible explanation for this result was that we cannot be certain that the adult wasps we randomly selected from the cages for RNA extraction were the ones we had observed to have fed on the dsRNA, or perhaps they might have been feeding on it and then transferring it to the larvae via trophallaxis thus diluting the amounts swallowed [46]. Therefore, when we pooled the data from different individuals in a cage for the gene expression analyses, it could have included wasps that did not fed on the dsRNA, modifying the overall expression results. This same 'dilution' effect might explain why there was no mortality recorded for any nests.

Protecting the dsRNA from degradation from saliva and gut nucleases works well in many insects [11, 47]. In our assays, we trialled two different methods of dsRNA protection. The first method, carbon quantum dots (CQD), did not seem to have worked as expected. CQD-protected dsRNA tended to increase expression of the targeted genes relative to the naked (unprotected) version of the same dsRNA. The second method, liposome encapsulation (lipofectamine), resulted in successful silencing of the DRE4 gene target. Lipofectamine transfection reagents have been considered as the 'gold standard' for the safe delivery of exogenous RNA into cells [48]. Future work with social wasps and dsRNA should incorporate this reagent to efficiently deliver dsRNA.

There is an extensive body of literature about the fortuitous application of RNAi in insects, as efficacy varies across insect taxa, genes examined, depending on the delivery mode, between laboratory or in the wild, and even between laboratories [2, 3]. We, initially, based our design

of dsRNA on a beetle study that ranked the 100 most lethal gene targets [27]. Some insects such as coleopterans exhibit both oral responses to RNAi (induced by dsRNA feeding) and systemic responses (the spreading of the RNAi effects from cell to cell) whereas in other insects such as *Drosophila* spp., the RNAi responses are cell-autonomous, i.e., restricted to the cell where the dsRNA is expressed or introduced [reviewed in 49]. In coleopterans, dsRNA is quickly processed to dsRNAs whereas in lepidopterans, the RNAi accumulates in the cells [50]. Lack of RNAi degradation and intracellular transport of dsRNA are the major factors responsible for reduced RNAi sensitivity in lepidopterans [50]. In some hymenopteran such as ecto- and endo-parasitic wasps, RNAi seems to have been applied to knockdown gene expression successfully [51]. We lack studies that could indicate potential reasons why RNAi might not be equally efficient in social wasps.

Overall, RNAi-mediated pest management could have several advantages compared to traditional control methods. It is considered as environmentally friendly, it has wide agronomic application, and is highly specific relative to pesticides [52]. However, it may not be a universal panacea for the management of all pest species. Even if paper wasp responses to dsRNA feeding consistently resulted in mortality, a particularly challenging problem would be the delivery method for such treatment, as there are not effective attractant or baits specific for paper wasps [14, 53], and feeding individual nests in the field is unrealistic.

## Supporting information

**S1 Table. List of genes whose expression was targeted in *Polistes dominula* foragers with double-stranded RNA (dsRNA) to assess mortality in this study.**
(PDF)

**S2 Table. Sequences for the 22 dsRNAs tested in *P. dominula* in this study and the qPCR primer pairs used to determine expression of the targeted genes.**
(PDF)

**S3 Table. Raw gene expression data, relative gene expression data, and survival curves data for the trials reported in this study.**
(XLSX)

**S1 Fig. Comparing the relative expression of calmodulin (CaM) gene present in the megacocktail with the GFP dsRNA and the sugar water control.** The qPCR primers unavoidably overlapped the dsRNA region for CaM. The RT-qPCR results clearly demonstrate CaM dsRNA consumption due to the apparent massive increase in CaM gene 'expression' due to the qPCR primers binding to the reverse-transcribed dsRNA rather than indicating a real increase in gene expression.
(PDF)

**S2 Fig. Survival of P. dominula forager wasps fed four different dsRNA gene targets at three different concentrations and in two presentations, naked dsRNA on the left or CQD-protected dsRNA shown on the right, for 10 days in captivity in the laboratory.**
(PDF)

## Acknowledgments

We thank Richard Toft for logistical assistance during wasp collection and Neil MacMillan for help collecting wasps.

## Author Contributions

**Conceptualization:** Mariana Bulgarella, James W. Baty, Philip J. Lester.

**Data curation:** Mariana Bulgarella, James W. Baty, Philip J. Lester.

**Formal analysis:** Mariana Bulgarella, James W. Baty, Philip J. Lester.

**Funding acquisition:** Philip J. Lester.

**Investigation:** Mariana Bulgarella, James W. Baty, Rose McGruddy.

**Methodology:** Mariana Bulgarella, James W. Baty, Philip J. Lester.

**Project administration:** Mariana Bulgarella, Philip J. Lester.

**Resources:** James W. Baty, Rose McGruddy, Philip J. Lester.

**Supervision:** Philip J. Lester.

**Validation:** Mariana Bulgarella, James W. Baty, Philip J. Lester.

**Visualization:** Mariana Bulgarella, Philip J. Lester.

**Writing – original draft:** Mariana Bulgarella.

**Writing – review & editing:** Mariana Bulgarella, James W. Baty, Rose McGruddy, Philip J. Lester.

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
