## [Decision Letter · Decision Letter 0]

25 Nov 2022

PONE-D-22-26973Gene silencing for invasive paper wasp management: synthesized dsRNA can modify gene expression but did not affect mortalityPLOS ONE

Dear Dr. Bulgarella,

Thank you for submitting your manuscript to PLOS ONE. After careful consideration, we feel that it has merit but does not fully meet PLOS ONE’s publication criteria as it currently stands. Therefore, we invite you to submit a revised version of the manuscript that addresses the points raised during the review process.

Two reviewers have praised the manuscript and had only minor concerns: The authors should take care to add all the necessary details for the dsRNA production and qPCR methods. I also agree with the reviewers that the figures could be improved. I do think it is useful to have the photographs in the main manuscript, but the current solution isn’t very nice. I wonder if the photos would work better as smaller inserts inside the bar graphs, or as separate figures? Also, there seems to be some inconsistency – no photograph with figure 2, but the corresponding picture seems to be in the supplement?

We look forward to receiving your revised manuscript.

Kind regards,

Volker Nehring

Academic Editor

PLOS ONE

Journal Requirements:

"We thank Richard Toft for logistical assistance during wasp collection and Neil MacMillan for help collecting wasps. This project was funded with a Smart Ideas Grant from the New Zealand Ministry of Business, Innovation and Employment number 61270-ENDSI-RSCHTUSTVIC."

"This study was funded by a Smart Ideas grant number PROP-61270-ENDSI-RSCHTRUSTVIC from the New Zealand Ministry of Business, Innovation and Employment, awarded to PJL. 

https://www.mbie.govt.nz/science-and-technology/science-and-innovation/funding-information-and-opportunities/investment-funds/endeavour-fund/application-and-assessment-information/

Reviewers' comments:

Reviewer's Responses to Questions

**Comments to the Author**

1. Is the manuscript technically sound, and do the data support the conclusions?

Reviewer #1: Yes

Reviewer #2: Yes

2. Has the statistical analysis been performed appropriately and rigorously? 

Reviewer #1: Yes

Reviewer #2: Yes

3. Have the authors made all data underlying the findings in their manuscript fully available?

Reviewer #1: No

Reviewer #2: Yes

4. Is the manuscript presented in an intelligible fashion and written in standard English?

Reviewer #1: Yes

Reviewer #2: Yes

5. Review Comments to the Author

Reviewer #1: Dear authors,

This paper examines the effects of RNAi on the wasp, Polistes dominula. The experiments are reasonable and well-written. I have made some comments, please see below.

It is strange that the results are shown here as Fig1; I think it would be better to separate the top and bottom of Fig.1 and just explain the methodology here using the bottom of Fig.1 (as a supplementary figure).

Was RNA extraction done with the whole body of the wasps?

As you cited, some genes are upregulated by RNAi (Fig.1; L.333-335). Could this overlap between the region where dsRNA was created and the region where it was quantified?

Related to the question above, the sequence of the primers used for dsRNA synthesis and qPCR should be provided, even if it is only supplementary information.

Minor comments

L. 32-35 These two sentences say almost the same thing.

L. 74 Delete “conjugation”.

L. 353 Delete “feeding wasps”.

L.488 DVSFN7 -> DVSNF7

L. 499 same -> similar

Reviewer #2: The authors did several different experiments attempting to identify feasible dsRNA candidates for pest control in the Polistes dominula invasion. They used both injected and oral delivery, with and without protective substances for the dsRNA, all in adult P. dominula. They found that both the injected and oral dsRNA (at least when protected with lipofectamine) were able to significantly reduce expression of some genes, but that none of them increased mortality.

I think this is a thorough and well-designed set of experiments. I appreciate the variety of approaches that were taken to investigate this question. I’m glad to see negative results like this coming through. This is the sort of thing that can easily be written off as a failed experiment, so it’s good to see the negative results being reported. I think that this article adds a very important contribution to the work being done with dsRNA in hymenoptera and to the work on the Polistes dominula invasion.

Major issues:

1. Please include the method used to produce dsRNA.

Minor issues:

1. Figure 1. The gene names trickling onto the picture makes it less clear. This would just benefit from some visual clean up.

2. Line 114: I think this should be “colonies were collected”

3. Given that previous studies have focused on targeting larval development with dsRNA and whole colonies were given dsRNA I find it disappointing that no data on larval development were reported and no measurements of gene expression were done in the larvae. I don’t think another experiment is required, but some discussion of this would be beneficial.

6. PLOS authors have the option to publish the peer review history of their article (what does this mean?). If published, this will include your full peer review and any attached files.

Reviewer #1: **Yes: **Yudai Nishide

Reviewer #2: **Yes: **Susan Weiner

---

## [Author Response · Author response to Decision Letter 0]

8 Dec 2022

Replies to comments from Reviewer 1

3. Have the authors made all data underlying the findings in their manuscript fully available?

Reviewer #1: No

We understand that Reviewer 1 would like to see more details about the dsRNA production and the qPCR methods. We have now expanded on both these sections as suggested.

5. Review Comments to the Author

Reviewer #1: Dear authors,

This paper examines the effects of RNAi on the wasp, Polistes dominula. The experiments are reasonable and well-written. I have made some comments, please see below.

Thank you very much for your time and useful comments. Much appreciated.

It is strange that the results are shown here as Fig1; I think it would be better to separate the top and bottom of Fig.1 and just explain the methodology here using the bottom of Fig.1 (as a supplementary figure).

Thank you for this suggestion. We have now separated the photographs from the graphs as the way in which we presented them together seemed distracting to the Reviewers and the Editor. There are now more figures in the manuscript but hopefully they make interpretation of the results easier. 

Was RNA extraction done with the whole body of the wasps?

Yes, it was. We have clarified this fact now in lines 297-299.

As you cited, some genes are upregulated by RNAi (Fig.1; L.333-335). Could this overlap between the region where dsRNA was created and the region where it was quantified?

This is only the case in three of the genes targets evaluated, calmodulin (CaM), DVSNF7 and RPB7 for which the primers necessarily overlapped the dsRNA region as stated in Table S2. We explained that this is the case for CaM in a paragraph in lines 392-398 that reads:

One benefit of this experiment was to confirm that wasps were feeding on the dsRNA. For the calmodulin (CaM) gene target, the qPCR primers unavoidably overlapped the dsRNA region. The RT-qPCR clearly demonstrated CaM dsRNA consumption as shown in the apparent massive increase in CaM gene ‘expression’ due to the qPCR primers binding to the reverse-transcribed dsRNA, rather than indicating an increase in gene expression (S1 Fig). This result also indicated that the lack of gene knockdown in other experiments is not due to the wasps not feeding on the dsRNA.

We also provided a separate figure for calmodulin ‘expression’ in the Supplemental material (S1 Fig). 

Related to the question above, the sequence of the primers used for dsRNA synthesis and qPCR should be provided, even if it is only supplementary information.

Reviewer 1 might have missed that all primer pairs used in our study are presented (and were in the original submission) in Supplementary Table 2. 

Minor comments

L. 32-35 These two sentences say almost the same thing.

You are right. We have combined the two sentences in one.

L. 74 Delete “conjugation”.

Done.

L. 353 Delete “feeding wasps”.

Done. 

L.488 DVSFN7 -> DVSNF7

Fixed, thank you!

L. 499 same -> similar

Changed.

Replies to comments from Reviewer 2

Reviewer #2: The authors did several different experiments attempting to identify feasible dsRNA candidates for pest control in the Polistes dominula invasion. They used both injected and oral delivery, with and without protective substances for the dsRNA, all in adult P. dominula. They found that both the injected and oral dsRNA (at least when protected with lipofectamine) were able to significantly reduce expression of some genes, but that none of them increased mortality.

I think this is a thorough and well-designed set of experiments. I appreciate the variety of approaches that were taken to investigate this question. I’m glad to see negative results like this coming through. This is the sort of thing that can easily be written off as a failed experiment, so it’s good to see the negative results being reported. I think that this article adds a very important contribution to the work being done with dsRNA in hymenoptera and to the work on the Polistes dominula invasion.

Thank you so much for such encouraging words. Indeed, these are not the results we expected when we planned to develop an alternative to insecticides as a control method to manage the paper wasp invasion in New Zealand. However, we also see value in communicating these ‘negative results’ to colleagues to avoid repetition and time consumption. Thank you very much for your time and suggestions. 

Major issues:

1. Please include the method used to produce dsRNA.

We have included the methods and details of dsRNA production in this revised version of the manuscript.

Minor issues:

1. Figure 1. The gene names trickling onto the picture makes it less clear. This would just benefit from some visual clean up.

Both reviewers and the editor suggested separating the photographs from the graphs so that has been done, thanks!

2. Line 114: I think this should be “colonies were collected”

Indeed. Thanks for finding this typo.

3. Given that previous studies have focused on targeting larval development with dsRNA and whole colonies were given dsRNA I find it disappointing that no data on larval development were reported and no measurements of gene expression were done in the larvae. I don’t think another experiment is required, but some discussion of this would be beneficial.

In our trials, we focused on adult wasps as in the field, should we provide dsRNA to wild wasps, we will be targeting the adult forager wasps to carry the dsRNA back to the colonies. We did collect data on larval mortality for the whole nest feeding trials but as most larvae emerged, we didn’t include these data in the manuscript. We have now included a short paragraph discussing this topic, lines 530-536 of the Discussion. 

Thank you again for your time and help, 

Mariana Bulgarella, on behalf of all co-authors

---

## [Editor Report · Decision Letter 1]

20 Dec 2022

Gene silencing for invasive paper wasp management: Synthesized dsRNA can modify gene expression but did not affect mortality

PONE-D-22-26973R1

Dear Dr. Bulgarella,

We’re pleased to inform you that your manuscript has been judged scientifically suitable for publication and will be formally accepted for publication once it meets all outstanding technical requirements.

Kind regards,

Volker Nehring

Academic Editor

PLOS ONE

Additional Editor Comments (optional):

It seems that the authors have made all info regarding the dsRNA available in the supplement. From what I can tell, the raw data obtained from the experiments (i.e. individual measures of gene expression levels, survival numbers for wasps) are still missing and I would suggest to add them as supplementary files.
---

## [Editor Report · Acceptance letter]

22 Dec 2022

PONE-D-22-26973R1 

Gene silencing for invasive paper wasp management: Synthesized dsRNA can modify gene expression but did not affect mortality 

Dear Dr. Bulgarella:

I'm pleased to inform you that your manuscript has been deemed suitable for publication in PLOS ONE. Congratulations! Your manuscript is now with our production department. 

Kind regards, 

on behalf of

Dr. Volker Nehring 

Academic Editor

PLOS ONE